REGISTERED REPORT PROTOCOL

# Efficacy and safety of manual acupuncture for the treatment of upper limb motor dysfunction after stroke: Protocol for a systematic review and meta-analysis

Di Cao[1‡], Xiaolin Zhang[2‡], Mingjun Liu[2,3], Qiguang Yang[1], Shuhong Gu[1], Tianjiao Gao[2], Lin Cong[2], Dehui Ma[2], Hongju Lin[1], Shaotao Chen[2,3]*

**1** Department of Rehabilitation, The Second Affiliated Hospital of Changchun University of Chinese Medicine (Changchun Hospital of Chinese Medicine), Changchun, China, **2** Changchun University of Chinese Medicine, Changchun, China, **3** Acupuncture and Massage Center of the Third Affiliated Hospital of Changchun University of Chinese Medicine, Changchun, China

‡ DC and XLZ contributed equally to this work as co-first authors
* ShaochenChen3600@163.com

## Abstract

### Introduction

The incidence of stroke sequelae among patients is as high as 70%–80%. Flexor spasm is the most common stroke sequela, presenting a heavy burden to the patients and their families. This study will evaluate the results of randomized controlled trials to determine the efficacy and safety of hand manipulation acupuncture for the treatment of upper limb motor dysfunction after stroke.

### Methods

Eight databases, including China National Knowledge Infrastructure, Chinese Scientific Journal Database, Cochrane Central Register of Controlled Trials, Embase, MEDLINE, PubMed, Wanfang Database, and Web of Science, will be searched using English and Chinese search strategies. In addition, manual retrieval of research papers, conference papers, ongoing experiments, and internal reports, among others, will supplement electronic retrieval. All eligible studies published on or before January 15, 2021 will be selected. To enhance the effectiveness of the study, only clinical randomized controlled trials related to the use of manual acupuncture for the treatment of upper limb motor dysfunction after stroke will be included.

### Analysis

The Fugl-Meyer upper extremity assessment will be the primary outcome measure, whereas the Wolf Motor Function Test, Modified Ashworth Scale, arm movement survey test table, and upper extremity freehand muscle strength assessment scores will be the secondary outcomes. Side effects and adverse events will be included as safety evaluations. To ensure the quality of the systematic evaluation, study selection, data extraction, and

**Data Availability Statement:** All relevant data from this study will be made available upon study completion.

**Funding:** This work was supported by the School construction project of National Administration of traditional Chinese medicine (Grant Number LPGZS22014-11, http://www.satcm.gov.cn/renjiaosi/zhengcewenjian/2018-03-24/1840.html), The Jilin Local Standard Construction Project (Grant Number: DBXM097-2020, http://scjg.jl.gov.cn/zw/gsgg/202105/t20210526_8082979.html) and the State Administration of Traditional Chinese Medicine of Jilin Province(Grant Number: zybz-zc-2020-004, http://jltcm.jl.gov.cn/tzgg/xgdt/202107/t20210706_8130847.html).

**Competing interests:** The authors have declared that no competing interests exist.

**Abbreviations:** 95% CIs, 95% confidence intervals; ARAT, Arm movement survey test table; FMA-UE, Fugl-Meyer assessment upper extremity; MAS, Modified Ashworth Score; MMT, upper extremity freehand muscle strength assessment; ULMD, Upper Limb Motor dysfunction; WMFT, Wolf motor function test.

quality assessment will be independently performed by two authors, and a third author will resolve any disagreement.

## Ethics and dissemination

This systematic review will evaluate the efficacy and safety of manual acupuncture for the treatment of upper limb motor dysfunction after stroke. Since all included data will be obtained from published articles, it does not require ethical approval and will be published in a peer-reviewed journal.

**INPLASY registration number**: INPLASY202110071.

## Introduction

Stroke, also known as cerebrovascular accident, is a common cardiovascular and cerebrovascular disease. It is classified into different forms, including ischemic and hemorrhagic stroke. According to the data released by the World Health Organization, one person has a stroke every 5 s, and about 15 million people experience a brain injury after stroke every year [1]. About 85% of stroke survivors have upper limb dysfunction, and more than 60% of them have persistent hand dysfunction and cannot live independently after treatment [2]. Studies have shown that the common clinical manifestations of upper limb motor dysfunction after stroke include muscle weakness, muscle spasm, and muscle tension changes. These symptoms can lead to difficulty in carrying out daily activities, such as reaching out, picking up, moving, and dressing, etc., thereby affecting a patient's quality of life [3, 4]. Fine motor dysfunction of the upper limbs after stroke, though a vital challenge, is a necessity of post-stroke treatment in order to reduce the stroke disability rate. Currently, the most commonly used rehabilitation therapies include physical factor therapy, exercise and occupational therapy, compensatory training, biofeedback, motor imagery therapy, contralateral C7 nerve root transfer, acupuncture, and massage [5, 6].

Studies have shown that manual acupuncture can stimulate the nerves of the human body, improve the motor status of patients, and have a positive effect on the recovery of patients and their independent actions [7–10]. According to the Chinese guidelines for the diagnosis and treatment of acute ischemic stroke published in 2018, acupuncture (level II recommendation, level B evidence) can be selected according to specific situations and the wishes of patients [11]. It is obvious that the curative effect of acupuncture in the treatment of upper limb motor dysfunction after stroke has been recognized by clinical experts.

Currently, the evidence levels of the available evidence-based research on manual acupuncture for the treatment of upper limb motor dysfunction after stroke are not high. This is because the existing studies have limitations such as small sample size, loosely implemented randomization, different degrees of blinding, use of nonstandard intervention measures in the control groups, and lack of evidence from high-quality randomized controlled trials (RCTs). Therefore, the aim of this study is to analyze the results of RCTs to ascertain the efficacy and safety of manual acupuncture for the treatment of upper limb motor dysfunction after stroke. This will allow for the provision of reliable evidence-based clarification of the efficacy of manual acupuncture for the treatment of upper limb motor dysfunction after stroke.

The proposed date for the completion of this study is October 15, 2021.

## Materials and methods

This protocol is based on the Preferred Reporting Items for Systematic Reviews and Meta-Analyses Protocols (PRISMA-P) guidelines [12] and the corresponding checklist. The systematic review protocol is registered in the INPLASY International Registry of Systematic Reviews (ID:INPLASY202110071; https://inplasy.com/?s=INPLASY202110071; DOI: 10.37766/inplasy2021.1.0071).

### Inclusion and exclusion criteria

**Types of participants.**   Patients' age will be between 18 and 75 years old. In line with China's 2015 diagnostic criteria for classifying cerebrovascular diseases, patients who had a first stroke confirmed using computed tomography or magnetic resonance imaging, a disease course longer than 1 month, but less than or equal to 6 months, and moderate to severe upper extremity and hand dysfunction that meet the standard criteria (Brunnstrom grade 2–4, Fugl Meyer Assessment [FMA] score < 20, improved Ashworth Spasm Scale score less than level 3 [13]) will be included, regardless of their sex and nationality. Exclusion criteria will include: 1) presence of other diseases resulting in serious cognitive or speech disorders 2) inability to understand and complete the therapist's instructions (Mini-Mental State Examination score < 21 points [14]) 3) history of drug or alcohol dependence 4) serious liver or kidney disease 5) other diseases that may affect brain structure and function; and 6) other mental disorders.

**Types of interventions.**   We will include studies in which the intervention group received manual acupuncture alone or in combination with routine rehabilitation treatment (manual therapy, exercise therapy, and electronic biofeedback, etc.), and in which the control group received only conventional rehabilitation treatment.

**Types of outcome measures.**   The primary outcome measure will be the FMA score. Secondary outcomes will include the Wolf Motor Function Test (WMFT), Modified Ashworth Scale(MAS), arm movement survey test table(ARAT), and upper extremity freehand muscle strength assessment scores(MMT) [15].

Relaxation periods and spasmodic periods will be analyzed in subgroup analyses.

**Types of studies.**   Randomized controlled clinical trials and quasi-RCTs will be included.

### Data sources

A structured and systematic literature search for eligible and relevant articles published on or before January 15, 2021 will be conducted. The following databases will be searched:China National Knowledge Infrastructure, Chinese Scientific Journal Database, Cochrane Central Register of Controlled Trials, Embase, MEDLINE, PubMed, Wanfang Database, and Web of Science. Selected studies will be published in either Chinese or English. The search terms include "cerebral apoplexy", "sequelae of stroke", "upper extremity motor dysfunction", "upper limb hemiplegia", "hemiplegia", "acupuncture", "clinical RCT".

### Searching other resources

TA manual search will mainly be used in searching for relevant studies. Details of the selection process are shown in a flow chart and the screening process is summarized in a flow diagram (Fig 1).

### Search strategy

The search strategy will be based on the Cochrane handbook guidelines (5.1.0) and will include keywords, such as "post-stroke", "after stroke", "acupuncture" or "manual acupuncture",

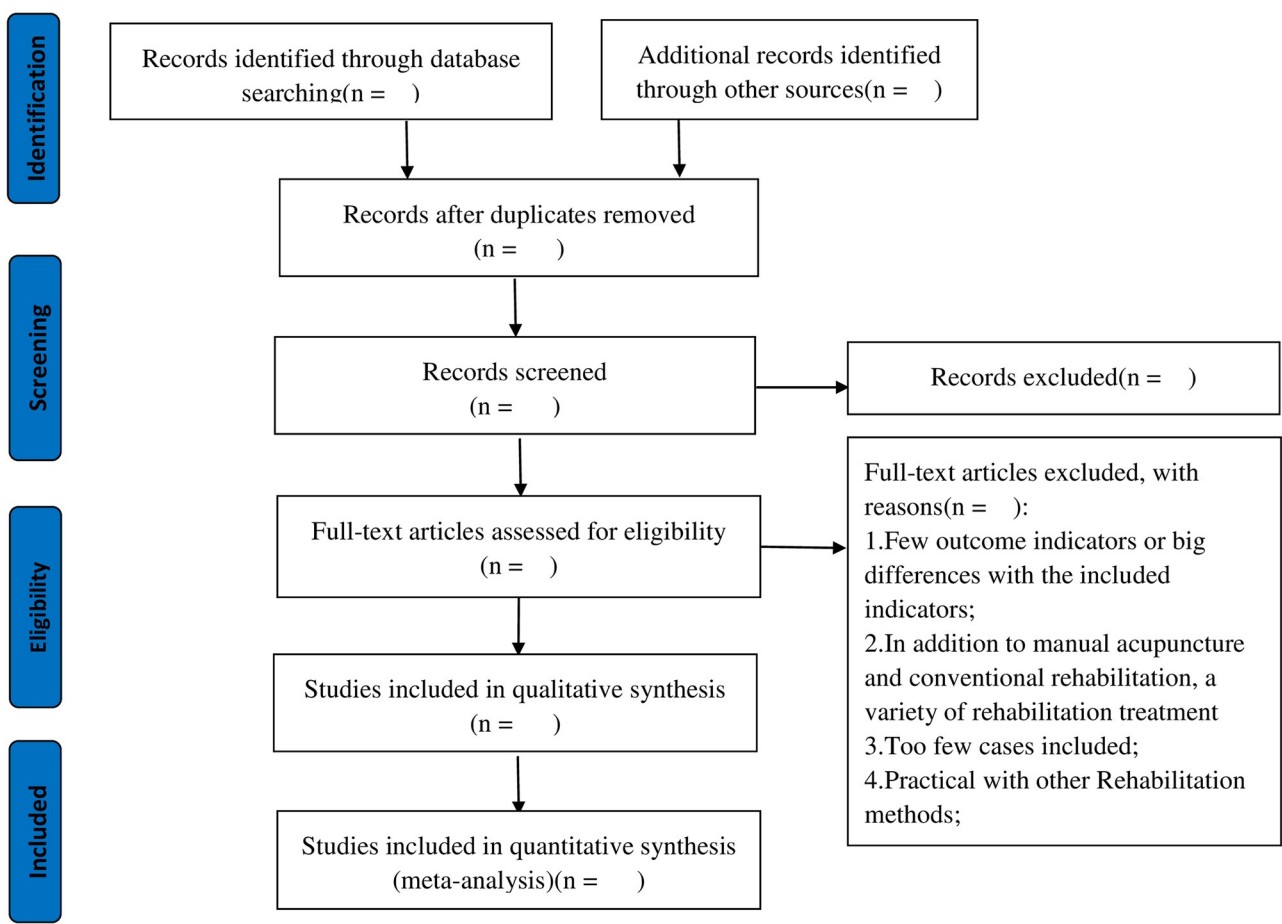

**Fig 1. PRISMA flow diagram of study and exclusion.**

"upper extremity motor dysfunction", and "clinical RCT". Subsequent searches will involve the use of Medical Subject Headings terms headings, in addition to keywords from the initial retrieval. Additional article searches will involve a review of the reference lists of relevant research articles. As an example, the search strategy for PubMed is summarized in Table 1.

## Selection of studies

Two researchers (GTJ and CL) will independently select the eligible literature according to the inclusion and exclusion criteria after reading their titles and abstracts. Subsequently, the full texts of the papers will be read and uncontrolled research, nonrandomized studies, and studies with inconsistent evaluation criteria or similar data will be excluded. If any differences occur during the screening, the third author (CST) would intervene.

## Data extraction and management

Two researchers (GTJ and CL) will use a predesigned data extraction table to extract the data of the included studies. The extracted data will include author, year, sample size, course of treatment, intervention measures, outcome indicators, adverse reactions, etc. The study selection procedure will be performed according to the PRISMA guidelines, which are presented in the flow diagram (Fig 1).

**Table 1. Search strategy for PubMed.**

| Number | Search terms |
|--------|--------------|
| 1 | acupuncture.ti,mesh. |
| 2 | manual acupuncture.ti,ab. |
| 3 | or 1–2 |
| 4 | post-stroke. ti,ab. |
| 5 | after stroke. ti,ab. |
| 6 | or 4–5 |
| 7 | Upper extremity motor dysfunction. ti,mesh. |
| 8 | Upper limb hemiplegia. ti,ab. |
| 9 | or 7–8 |
| 10 | randomised controlled trial.pt. |
| 11 | Controlled clinical trial.pt. |
| 12 | randomised.ab. |
| 13 | Randomly.ab. |
| 14 | trial.ab. |
| 15 | or 10–15 |
| 16 | 3 and 6 and 9 and 15 |

## Statistical analysis

We will use SAS (SAS Institute, Cary, NC) and Stata (StataCorp, College Station, TX) to analyze the standard deviation, standard error and mean of each group. When we encounter literature with missing data, we will try to contact the author. If we are unable to get full data, then that study will not be included. We will use the Review Manager 5.3 software provided by the Cochrane Collaborative Network for statistical analysis [16]. For continuous variables, the means and standard deviations of each study will be obtained and pooled as a mean difference or a standardized mean difference with a 95% confidence interval. The statistical heterogeneity of the included clinical RCTs will be analyzed. The $I^2$ test will be used to test for heterogeneity. When $I^2$ is <50% or P>0.05, it indicates that there is no statistical heterogeneity between studies [17]. A fixed-effects model will be selected to combine the effect amount; otherwise, a random effects model will be adopted.

## Methodological assessment of quality

The qualities of the included studies will be evaluated using the risk of bias table proposed by the Cochrane Collaborative Network [17]. The risk table includes six items: random sequence generation mode, whether to use allocation concealment, whether to blind the subjects and intervention providers, whether to blind the result evaluators, whether the result data are complete, whether to select the result report, and other bias sources. The criteria used to assess the risk of bias are "low risk", "high risk", and "unclear" [17]. Two evaluators will independently evaluate the methodological qualities of the studies. In cases of disagreement, the third author would intervene.

## Assessment of heterogeneity

If there is no significant heterogeneity ($I^2$<50%)between studies, the fixed-effects model will be used for evaluation. If there is significant heterogeneity ($I^2$>50%), the random-effects model will be used for evaluation [17]. Sensitivity analysis or subgroup analysis will then be conducted as required to explain the heterogeneity.

## Subgroup analysis

If possible, subgroups will be analyzed according to relaxation and spasm or different acupuncture manipulation.

## Sensitivity analysis

When possible, we will perform sensitivity analysis to explore the effects of a trial's risk of bias on primary outcomes. Lower quality trials will be excluded from these analyses and the meta-analyses will be repeated according to sample size and insufficient data to assess quality and robustness when significant statistical heterogeneity arises.

## Assessment of publication bias

If more than 10 trials meet the study criteria, we will use Review Manager 5.3 software to draw and analyze a funnel chart and use the funnel chart to evaluate the potential publication bias.

## Grading the quality of evidence

The Grading of Recommendations Assessment, Development and Evaluation (GRADE) approach [18] is recommended for the analysis of the level of evidence.

# Discussion

Stroke is a major disease that threatens human health. It has high rates of incidence, mortality, disability, recurrence, and economic burden. Data reported in the 2015 China Stroke Prevention report indicates that stroke has become China's foremost cause of death, resulting in an economic burden of up to 40 billion Yuan [19]. About 65%–80% of new stroke patients have upper limb dysfunction every year [20]. The recovery of upper limb motor function is slow and difficult, and it seriously affects the life and work of patients. It is a challenging and controversial point of rehabilitation treatment. Currently, the methods for treatment of upper limb motor dysfunction after stroke include the use of a robotic upper limb, contralateral controlled functional electrical stimulation, brain computer interface, and mirror therapy, among others [21–24]. However, these methods have some limitations in terms of clinical efficacy, popularity, and safety.

## Acupuncture has become an important method for rehabilitation of upper limb motor dysfunction after stroke

Acupuncture for the treatment of stroke has unique advantages [25]. With the increasingly widespread use of acupuncture globally, the mechanism of acupuncture in functional recovery after stroke has become a research hotspot. Studies [25–27] have demonstrated that acupuncture can promote the proliferation of central nervous system cells in cerebral ischemia-reperfusion injury; promote angiogenesis in infarcted areas; regulate local blood flow by vasoactive mediators, inhibit cell apoptosis; facilitate the regulation of neurochemicals, such as neurotransmitters, antioxidants, inflammatory related factors, and neurotrophic factors, and activate specific motor functional areas like the cortex. Acupuncture treatment has been widely accepted because of its simplicity, convenience, fewer side effects, and exact curative effect. It is a popular trend to combine acupuncture with modern rehabilitation technology [28].

## Clinical effect of manual acupuncture in the treatment of upper limb motor dysfunction after stroke is better than that of electroacupuncture

There are many reports on the treatment of upper limb motor dysfunction after stroke by manual acupuncture and electroacupuncture. While the electroacupuncture procedure is simple than manual acupuncture, we found that the effect of manual acupuncture is better through clinical research. Because the upper limb motor dysfunction in ischemic stroke occurs in the six stages of Brunnstrom motor function recovery [29], electroacupuncture treatment can decrease the patient's muscle strength lower in the flaccid period and increase it abnormally in the spasmodic period. Manual acupuncture is better during the six Brunnstrom stages because it can be changed according to what is needed.

## FMA-UE and other outcome indicators are closely related to upper limb motor dysfunction after stroke

The FMA-UE is a motor function assessment method designed by Fugl Meyer AR and colleagues for hemiplegic stroke patients with upper limb motor dysfunction according to Brunnstrom's theoretical framework [30]. It covers five areas of motion, sensation, balance, joint range of motion and pain, with a total of 113 assessment items and a full score of 226. It is the most commonly used scale to evaluate the therapeutic effect of upper limb motor function after stroke, and it is often used as the "gold standard" to test the validity of other scales [31, 32]. It can effectively evaluate the motor function and coordination of the shoulder, elbow, forearm, hand, and wrist. The evaluation content is more detailed, but the evaluation of fine finger fine activity is lacking [33]. On the other hand, ARAT can better evaluate the motor function of the distal end of the upper limb, including the evaluation of fine finger activity of stroke patients at various stages, and it is easy to operate, time-consuming, and easy to apply in clinical and scientific research [34, 35]. The WMFT can quantitatively evaluate the upper limb movement ability of patients through the timing of single joint movement, multi joint movement, functional activity, and the evaluation of movement quality. It has good validity and reliability. It is the most commonly used scale around the world to evaluate the improvement of upper limb function by compulsory exercise therapy [36]. The advantage of this method is that it does not only evaluate the quality of the work activity, but also measures the time of the operation activity, and has a good standard validity with the upper limb of FMAS [37]. Different from FMA-UE, WMFT can reflect many functions. The effect of task training on upper limb motor function. It can not only evaluate the degree of upper limb motor function injury, but can also evaluate the therapeutic effect of different interventions on upper limb injury. The operation method of elbow flexion spasm is more detailed, which is easy to achieve the evaluation standard. The reliability of the study is partly reliant on the evaluator's operation experience, repeated tests and discussions. It is mainly used in the auxiliary evaluation of elbow flexion spasm in the rehabilitation process of patients with upper limb motor dysfunction after stroke [38]. Combined with these evaluation methods, they can be evaluated from multiple angles and play a complementary role, making the evaluation results more reliable. The mental state of stroke patients is also very important for the recovery of dysfunction. There is currently no commonly used scale to evaluate the mental state of stroke patients in the literature.

There are currently no data on the systematic reviews or meta-analyses of the therapeutic effect of manual acupuncture on upper limb motor dysfunction after stroke. Therefore, we plan to conduct a systematic review and meta-analysis to provide high-quality evidence of the efficacy of manual acupuncture for upper limb motor dysfunction after stroke.

## Limitations

As manual acupuncture is a holistic method and follows the concept of "treatment based on syndrome differentiation," different patients and different acupuncturists will produce different therapeutic effects, leading to a certain level of heterogeneity. In addition, different types of manual acupuncture, including reinforcing and reducing, flat reinforcing and reducing, and other manipulations, also produce different therapeutic effects.

Second, the following drawbacks cannot be avoided during literature collection: the quality of literature not being high enough, the exclusion of studies not published in Chinese or English, and the sample size of the included literature not being large enough.

## Supporting information

**S1 Checklist. PRISMA-P (Preferred Reporting Items for Systematic review and Meta-Analysis Protocols) 2015 checklist: Recommended items to address in a systematic review protocol**<sup>*</sup>.
(DOC)

## Author Contributions

**Conceptualization:** Di Cao, Xiaolin Zhang, Shaotao Chen.

**Data curation:** Xiaolin Zhang, Tianjiao Gao, Lin Cong.

**Formal analysis:** Xiaolin Zhang.

**Funding acquisition:** Mingjun Liu.

**Methodology:** Qiguang Yang, Shuhong Gu.

**Software:** Tianjiao Gao, Lin Cong, Dehui Ma, Hongju Lin.

**Supervision:** Shaotao Chen.

**Writing – original draft:** Di Cao, Xiaolin Zhang.

**Writing – review & editing:** Di Cao, Xiaolin Zhang, Mingjun Liu.

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
