## [Decision Letter · Decision Letter 0]

10 May 2021

PONE-D-21-03553

Efficacy and safety of manual acupuncture in the treatment of Upper Limb Motor dysfunction after stroke: A protocol for systematic review and meta-analysis

PLOS ONE

Dear Dr. Chen,

Thank you for submitting your manuscript to PLOS ONE. After careful consideration, we feel that it has merit but does not fully meet PLOS ONE’s publication criteria as it currently stands. Therefore, we invite you to submit a revised version of the manuscript that addresses the points raised during the review process.

Please be aware that all queires raised by the reviewers need to be thoroughly addressed. The manuscript must meet all of PLOS ONE’s publication criteria.

Specifically the manuscript needs to be revised regarding its very poor language, its recurring repetitions throughout the manuscript partly copying whole paragraphs from other manuscripts. Legibility, intelligebility and novelty are required. The research must meet all applicable standards for research integrity.

We look forward to receiving your revised manuscript.

Kind regards,

Johannes Fleckenstein

Academic Editor

PLOS ONE

Journal Requirements:

"This work was supported by the School construction project of State

Administration of Chinese medicine (Grant Number LPGZS2014-11), The Jilin Local

Standard Construction Project (Grant Number: DBXM097-2020) and

Standardization construction project of Jilin Provincial Administration of Chinese

medicine (zybz-zc-2020-004)."

"he authors have declared that no competing interests exist."

6. Please include your tables as part of your main manuscript and remove the individual files. Please note that supplementary tables should remain uploaded as separate "supporting information" files.

8. Thank you for submitting the above manuscript to PLOS ONE. During our internal evaluation of the manuscript, we found significant text overlap between your submission and the following previously published works, some of which you are an author.

https://journals.lww.com/md-journal/Fulltext/2020/07240/Effectiveness_comparisons_of_catgut_implantation.68.aspx

https://bmjopen.bmj.com/content/11/1/e042383

Please revise the manuscript to rephrase the duplicated text, cite your sources, and provide details as to how the current manuscript advances on previous work. Please note that further consideration is dependent on the submission of a manuscript that addresses these concerns about the overlap in text with published work.

9. We suggest you thoroughly copyedit your manuscript for language usage, spelling, and grammar. If you do not know anyone who can help you do this, you may wish to consider employing a professional scientific editing service.  

Reviewers' comments:

Reviewer's Responses to Questions

**Comments to the Author**

1. Does the manuscript provide a valid rationale for the proposed study, with clearly identified and justified research questions?

Reviewer #1: Yes

Reviewer #2: Yes

2. Is the protocol technically sound and planned in a manner that will lead to a meaningful outcome and allow testing the stated hypotheses?

Reviewer #1: Yes

Reviewer #2: Partly

3. Is the methodology feasible and described in sufficient detail to allow the work to be replicable?

Reviewer #1: Yes

Reviewer #2: No

4. Have the authors described where all data underlying the findings will be made available when the study is complete?

Reviewer #1: Yes

Reviewer #2: Yes

5. Is the manuscript presented in an intelligible fashion and written in standard English?

Reviewer #1: No

Reviewer #2: No

6. Review Comments to the Author

You may also provide optional suggestions and comments to authors that they might find helpful in planning their study.

Reviewer #1: There are a number of English errors in this document. Examples are:

INTRO:

…these damages can lead to daily activities (such as reaching out, picking up, moving, dressing, etc.) obstacles…

Therefore, as an important part of reducing stroke disability rate, upper limb rehabilitation after stroke is the difficulty of post-stroke treatment.

Part 2 Starting with 2.2. Data source

Past, present, and future tenses are comingled within sentences and paragraphs.

2.8.Subgroup analysis If possible,Subgroups will analyzed according to relaxation and spasm or different acupuncture manipulation

Discussion:

Chinese medicine acupuncture treatment of stroke has its unique advantages，In recent years, with the application of acupuncture in the world more and more widely, the mechanism of acupuncture in functional recovery after stroke has become a research hotspot. [sentence structure, run-on sentence]

Editing is needed before this manuscript can be published

Reviewer #2: General comment:

The manuscript needs extensive linguistic correction. There are so many typos and punctuation errors. Some words and sentences are hardly understandable (e.g. “The fine dysfunction of the upper limbs, …”, “bind method”, “.. and Spasmodic periowill be…, MEDLIN, …high Gao Zhican rate… etc.). The tenses used are inconsistent throughout the methods section and should be carefully reviewed. Some sections are simply copied and repeated in different sections of the manuscript.

INTRODUCTION:

-Please ascertain that this description of the purpose statement is concise and accurate: “Therefore, in order to better study the efficacy and safety of manual acupuncture in the treatment of upper limb motor dysfunction after stroke, this study will be combined with the published clinical randomized controlled trials for analysis in order to provide reliable evidence-based basis for the efficacy of manual acupuncture in the treatment of upper limb motor dysfunction after stroke.”

- The described problem of the lack of high-quality RCTs (please provide some references) is not necessarily solved by conducting a systematic review and meta-analysis. This point should rather be addressed in the discussion section.

METHODS:

-Some parts (e.g. 2.1.1. Types of participants, 2.8. Sensitivity analysis, etc.) of this manuscript are almost identical with a recent publication of the same study group (Zhang X, Cao D, Liu J, et al. Effectiveness and safety of brain-computer interface technology in the treatment of poststroke motor disorders: a protocol for systematic review and meta-analysis. BMJ Open 2021;11:e042383. doi:10.1136/bmjopen-2020-042383). As it is mainly “copy paste” it should be reviewed and rewritten carefully or at least referenced throughout.

- Please list the databases used alphabetically

- Search terms are repeated in chapter 2.2 und search strategy.

- Reference on chapter 2.6 is missing.

- Chapter 2.11 “It is recommended to use…”. Are the authors going to use GRADE or not?

STATISTICAL ANALYSIS

-How will the authors deal with missing data? Please add the details.

-To the best of my knowledge RevMan does not offer any statistical tools (e.g., Egger’s test, I2-Test etc.). Which data analysis tools will the authors use to perform statistical tests? Please add details.

DISCUSSION:

- Please discuss and put in relation pro and cons between already established therapeutic options and acupuncture treatment.

- The authors mentioned that “at present, acupuncture treatment has been widely accepted because of its simple operation, convenience, less side effects and exact curative effect…” Please provide a rationale and add references.

- In my personal experience of treating stroke sequelae with acupuncture in China, I have seen electroacupuncture used almost exclusively. Why is this treatment option not included in the review but only manual acupuncture? Please provide a rationale or address this topic in the discussion section.

TABLES/SUPPL. TABLES:

- Search strategy for PubMed – Number 15 (or 10-15) should be corrected.

- Please provide an explanation for the used abbreviations e.g. ti,ab

ABSTRACT:

Please apply the revisions to the abstract as well.

7. PLOS authors have the option to publish the peer review history of their article (what does this mean?). If published, this will include your full peer review and any attached files.

Reviewer #1: **Yes: **Jennifer Brett, ND, L.Ac.

Reviewer #2: **Yes: **Jan Valentini

---

## [Author Response · Author response to Decision Letter 0]

2 Jun 2021

Dear Editors and Reviewers:

Thank you for your letter and for the reviewers’ comments concerning our manuscript entitled “Efficacy and safety of manual acupuncture in the treatment of Upper Limb Motor dysfunction after stroke: A protocol for systematic review and meta-analysis” (ID: PONE-D-21-03553). Those comments are all valuable and very helpful for revising and improving our paper, as well as the important guiding significance to our research. We have studied comments carefully and have made the correction which we hope meet with approval. The revised portion is marked in red in the paper. The main corrections in the paper and the responses to the reviewer’s comments are as flowing:

Responds to the reviewer’s comments:

Reviewer #1:

1. Response to comment: (……There are some mistakes in English in this document……)

Response: Because English, as a second language, does suggest to you that there are some mistakes in the article. This time, we have carefully revised and checked sentence by sentence, and even applied for professional translation assistance from the translation and editing agency.

Reviewer #2:

1. Response to comment: (Please ascertain that this description of the purpose statement is concise and accurate: “Therefore,...... stroke.”)

Response: ××××××

2. Response to comment: (……The described problem of the lack of high-quality RCTs (please provide some references) is not necessarily solved by conducting a systematic review and meta-analysis. This point should rather be addressed in the discussion section.……)

Response: ××××××

3.Response to comment: (……Some parts (e.g. 2.1.1. Types of participants, 2.8. Sensitivity analysis, etc.)……)

Response: We have completed the modification according to the suggestions. We mistakenly thought that the repetitive part is that we can quote the previous articles published in BMJopen.

4.Response to comment: (……- Please list the databases used alphabetically……)

Response:According to the suggestion, we have listed the databases used in this paper in alphabetical order.

5.Response to comment: (……STATISTICAL ANALYSIS

-How will ……perform statistical tests? Please add detail……)

Response:We will use Statistics Analysis System and Stata to analyze the Standard Deviation, Standard Error and Mean of each group.

When we encounter the literature with missing data, we will try to contact the author. If we can't get the full data through contact, we will exclude the author's full data.

6.Response to comment: (…… In my personal ……Please provide a rationale or address this topic in the discussion section……)

Response:Because the upper limb motor dysfunction of ischemic stroke occurs in the six stages of Brunnstrom motor function recovery, sometimes electroacupuncture treatment is easy to make the patient's muscle strength lower in the flaccid period and increase abnormally in the spasmodic period. And manual acupuncture can avoid this problem very well. At the same time, it can achieve better curative effect by changing the manual in six different periods.

---

## [Decision Letter · Decision Letter 1]

1 Jul 2021

PONE-D-21-03553R1

Efficacy and safety of manual acupuncture for the treatment of upper limb motor dysfunction after stroke: Protocol for a systematic review and meta-analysis

PLOS ONE

Dear Dr. Chen,

Thank you for submitting your manuscript to PLOS ONE. After careful consideration, we feel that it has merit but does not fully meet PLOS ONE’s publication criteria as it currently stands. Therefore, we invite you to submit a revised version of the manuscript that addresses the points raised during the review process.

The prerequisites for a successful re-submission are:

1) major improvements in language and style (please consult a native speaker)

2) a thorough revision of the discussion section

We look forward to receiving your revised manuscript.

Kind regards,

Johannes Fleckenstein

Academic Editor

PLOS ONE

Reviewers' comments:

Reviewer's Responses to Questions

**Comments to the Author**

1. Does the manuscript provide a valid rationale for the proposed study, with clearly identified and justified research questions?

Reviewer #1: Yes

Reviewer #2: Yes

2. Is the protocol technically sound and planned in a manner that will lead to a meaningful outcome and allow testing the stated hypotheses?

Reviewer #1: Yes

Reviewer #2: Yes

3. Is the methodology feasible and described in sufficient detail to allow the work to be replicable?

Reviewer #1: Yes

Reviewer #2: Yes

4. Have the authors described where all data underlying the findings will be made available when the study is complete?

Reviewer #1: Yes

Reviewer #2: Yes

5. Is the manuscript presented in an intelligible fashion and written in standard English?

Reviewer #1: Yes

Reviewer #2: No

6. Review Comments to the Author

You may also provide optional suggestions and comments to authors that they might find helpful in planning their study.

Reviewer #1: Revised document corrected the English language errors of the original submission.

Reviewer #2: The authors did a good job in revising the manuscript. Although many typos and language punctuation errors were corrected, there are still some remaining in the manuscripts (e.g. high Gao Zhican rate .. the collation of literature, etc.). Please check once again for spelling and language errors.

Some references are still missing in the manuscript. Please check whether you have provided sufficient references for your statements. E.g.:

- Methods: -pg. 36: The qualities of the included studies will be evaluated using the risk of bias table proposed by Cochrane collaborative network.

- Discussion pg. 28: Acupuncture for the treatment of stroke has unique advantages. With the increasingly widespread use of acupuncture globally, the mechanism of acupuncture in functional recovery after stroke has become a research hotspot.

Or

It is a popular trend to combine acupuncture with modern rehabilitation technology.

While some of the reviewers' comments from the first submission were addressed in the revision, others were neither addressed nor mentioned in the response to the reviewers. A point-by-point response would be welcome and could solve this problem.

7. PLOS authors have the option to publish the peer review history of their article (what does this mean?). If published, this will include your full peer review and any attached files.

Reviewer #1: **Yes: **Jennifer Brett

Reviewer #2: **Yes: **Jan Valentini, MD

---

## [Author Response · Author response to Decision Letter 1]

10 Jul 2021

Dear Prof. Fleckenstein and Reviewers:

Thank you for your letter and for the reviewers’ comments concerning our manuscript entitled “Efficacy and safety of manual acupuncture in the treatment of Upper Limb Motor dysfunction after stroke: A protocol for systematic review and meta-analysis” (PONE-D-21-03553R1). Those comments are all valuable and very helpful for revising and improving our paper, as well as the important guiding significance to our research. We focused on the two main problems that the editor helped us to summarize. First of all, we added a large number of relevant information according to the full text, consulted relevant experts, and completed the discussion part again. Then, we have made great improvements in the language and style of the full text, and invited "editing" organizations to help with native language services.

The revised portion is marked in red in the paper. The main corrections in the paper and the responses to the reviewer’s comments are as flowing:

Responds to the reviewer’s comments:

Reviewer #2:

1. Response to comment: (there are still some remaining in the manuscripts (e.g. high Gao Zhican rate .. the collation of literature, etc.). Please check once again for spelling and language errors.)

Response: First of all, we are sorry that we have not completely checked out spelling and language errors. This time, we have carefully checked spelling and language errors for many times to minimize errors.

2. Response to comment: (Some references are still missing in the manuscript)

Response:Thank you very much for helping us find some references missing. After careful proofreading, we added 9 references in the right place.

We tried our best to improve the manuscript and made some changes in the manuscript. These changes will not influence the content and framework of the paper. And here we did not list the changes but marked in red in revised paper.

We appreciate for Editors and Reviewers’ warm work earnestly and hope that the correction will meet with approval.

Once again, thank you very much for your comments and suggestions.

Authors:Di Cao, Xiaolin Zhang, Mingjun Liu, Qiguang Yang, Shuhong Gu, Tianjiao Gao, Lin Cong, Dehui Ma, Hongju Lin, Shaotao Chen.

E-mail:Shaochenchen3600@163.com.

Institutional: Changchun University of Chinese medicine

---

## [Decision Letter · Decision Letter 2]

7 Aug 2021

PONE-D-21-03553R2

Efficacy and safety of manual acupuncture for  the treatment of u pper l imb m otor dysfunction after stroke: P rotocol fo r a  systematic review and meta-analysis

PLOS ONE

Dear Dr. Chen,

Thank you for submitting your manuscript to PLOS ONE. After careful consideration, we feel that it has merit but does not fully meet PLOS ONE’s publication criteria as it currently stands. Therefore, we invite you to submit a revised version of the manuscript that addresses the points raised during the review process.

We acknowledge that the authors sought help with linguistic proofreading. Whereas grammar and style improved, in agreement with reviewer #2,  the manuscript still needs revisions in accuracy and punctuation.

We look forward to receiving your revised manuscript.

Kind regards,

Johannes Fleckenstein

Academic Editor

PLOS ONE

Journal Requirements:

Reviewers' comments:

Reviewer's Responses to Questions

**Comments to the Author**

1. Does the manuscript provide a valid rationale for the proposed study, with clearly identified and justified research questions?

Reviewer #2: Yes

2. Is the protocol technically sound and planned in a manner that will lead to a meaningful outcome and allow testing the stated hypotheses?

Reviewer #2: Yes

3. Is the methodology feasible and described in sufficient detail to allow the work to be replicable?

Reviewer #2: Yes

4. Have the authors described where all data underlying the findings will be made available when the study is complete?

Reviewer #2: Yes

5. Is the manuscript presented in an intelligible fashion and written in standard English?

Reviewer #2: No

6. Review Comments to the Author

You may also provide optional suggestions and comments to authors that they might find helpful in planning their study.

Reviewer #2: The authors have again done a good job in revising the manuscript. However, there are still some punctuation and grammatical errors (e.g. Modified Ashworth Scale(MAS),, arm movement survey test table(ARAT),and upper extremity freehand muscle strength assessment scores(MMT).[15] OR While the electroacupuncture procedure is more simple than manual acupuncture, but we found that the effect of manual acupuncture is better through clinical research). Please also provide a reference for this statement.

I would suggest seeking the help of a language editor again after the revision.

7. PLOS authors have the option to publish the peer review history of their article (what does this mean?). If published, this will include your full peer review and any attached files.

Reviewer #2: **Yes: **Jan Valentini, MD

---

## [Author Response · Author response to Decision Letter 2]

6 Oct 2021

Dear Prof. Fleckenstein and Reviewers:

Thank you for your letter and for the reviewers’ comments concerning our manuscript entitled “Efficacy and safety of manual acupuncture in the treatment of Upper Limb Motor dysfunction after stroke: A protocol for systematic review and meta-analysis” (PONE-D-21-03553R1). Those comments are all valuable and very helpful for revising and improving our paper, as well as the important guiding significance to our research. 

English is not our mother tongue and may still be not particularly accurate in some expressions, so there are still problems in language expression after the previous three modifications. This time, we specially invited a professional language polishing organization "Yideji" to help with native language services.

The revised portion is marked in red in the paper.

In addition, This is our first time to contribute to our "PLoS One". We found that"Funding Statemen" is different from other magazines. Our actual funding information is as follows:This work was supported by the School construction project of National Administration of traditional Chinese medicine (Grant Number LPGZS22014-11，http://www.satcm.gov.cn/renjiaosi/zhengcewenjian/2018-03-24/1840.html) , The Jilin Local Standard Construction Project (Grant Number: DBXM097-2020，http://scjg.jl.gov.cn/zw/gsgg/202105/t20210526_8082979.html) and the State Administration of Traditional Chinese Medicine of Jilin Province(Grant Number: zybz-zc-2020-004，http://jltcm.jl.gov.cn/tzgg/xgdt/202107/t20210706_8130847.html). Please help to modify it.

We tried our best to improve the manuscript and made some changes in the manuscript. These changes will not influence the content and framework of the paper. And here we did not list the changes but marked in red in revised paper.

We appreciate for Editors and Reviewers’ warm work earnestly and hope that the correction will meet with approval.

Once again, thank you very much for your comments and suggestions.

Authors:Di Cao, Xiaolin Zhang, Mingjun Liu, Qiguang Yang, Shuhong Gu, Tianjiao Gao, Lin Cong, Dehui Ma, Hongju Lin, Shaotao Chen.

E-mail:Shaochenchen3600@163.com.

Institutional: Changchun University of Chinese medicine

---

## [Editor Report · Decision Letter 3]

11 Oct 2021

Efficacy and safety of manual acupuncture for the treatment of upper limb motor dysfunction after stroke: Protocol for a systematic review and meta-analysis

PONE-D-21-03553R3

Dear Dr. Chen,

We’re pleased to inform you that your manuscript has been judged scientifically suitable for publication and will be formally accepted for publication once it meets all outstanding technical requirements.

Kind regards,

Johannes Fleckenstein

Academic Editor

PLOS ONE
---

## [Editor Report · Acceptance letter]

25 Oct 2021

PONE-D-21-03553R3 

Efficacy and safety of manual acupuncture for the treatment of upper limb motor dysfunction after stroke: Protocol for a systematic review and meta-analysis 

Dear Dr. Chen:

I'm pleased to inform you that your manuscript has been deemed suitable for publication in PLOS ONE. Congratulations! Your manuscript is now with our production department. 

Kind regards, 

on behalf of

Priv.-Doz. Dr. Johannes Fleckenstein 

Academic Editor

PLOS ONE